# Constitutionalising the Right to Water in Kenya and Slovenia: Domestic Drivers, Opportunity Structures, and Transnational Norm Entrepreneurs

**Mathea Loen * and Siri Gloppen**

The CMI-UiB Centre on Law & Social Transformation, Department of Administration and Organization Theory, University of Bergen, 5007 Bergen, Norway; siri.gloppen@uib.no

* Correspondence: mathea.loen@uib.no; Tel.: +47-913-35-843

**Abstract:** The international norm development that in 2010 culminated with the UN Resolution on the Human Right to Water and Sanitation changed international law. To what extent did this influence the parallel legal developments evident in many national constitutions across the globe? This article analyses the mobilisation for a constitutional right to water and sanitation in Kenya and Slovenia, identifying the main national and transnational actors involved and assessing their significance for the processes of constitutionalising the right. By analysing two very different cases, tracing their constitutionalisation processes through analysis of archival material, the article provides multifaceted insights into processes of norm diffusion from international norm entrepreneurs to the national level and the agency of domestic actors and their opportunity structures. We find that although the outcomes of the processes in Kenya and Slovenia are similar in that both constitutions contain articles securing the right to water, the framing of the right differs. Furthermore, we conclude that while there is involvement of international actors in both cases, domestic pro-water activists and their normative and political opportunity structures are more important for understanding the successful constitutionalisation of the right to water and differences in the framing of the right.

**Keywords:** human rights to water and sanitation; water access; constitutionalisation; norm diffusion; opportunity structures

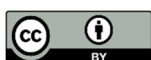

## 1. Introduction

In the past 20 years, there has been an acceleration in domestic constitutionalisation of the right to water following the adoption of the UN Resolution on the Human Right to Water and Sanitation in 2010. This indicates a diffusion of the international norm development to the national level [1]. In this article, we look more carefully at the dynamics at play to understand if and how such diffusion from the global level is happening or whether the domestic norm development is more autochthonous and driven by local actors. We probe this by identifying the main national and transnational actors involved in the processes of constitutionalising the right to water in Kenya and Slovenia and analysing how they worked jointly and separately to push for constitutionalisation and to influence the content of the norm in each case. Thus, our central research question is as follows: How does the right to secure, adequate access to water become constitutionalised, and how do processes of global norm development play into the social and political construction of the right at the domestic level? This is explored through two sub-questions: firstly, how do local activists mobilise around the right to water in different domestic contexts, and secondly, how do transnational norm entrepreneurs interact with local actors to influence the constitutionalisation of the right through mechanisms of norm diffusion?

Following the introduction, the remaining article is divided into six parts. Section two presents the theoretical lens used in the article, which combines theoretical work on international norm entrepreneurs, norm diffusion [2,3], opportunity structures [4], and water governance [5–7]. The third section presents the methodology and data material that will be used to trace the domestic processes of norm development and the interaction of local actors and transnational norm entrepreneurs. In section four, we briefly outline the trajectory of the human right to water as a human right norm from the International Convention on Economic Social and Cultural Rights (ICESCR, 1966) and the first discussion of water as a human right at the United Nations Water Conference in Mar Del Plata in 1977 to the adoption of the 2010 UN resolution. Based on this timeline, we present the main norm entrepreneurs involved in this development. Sections five and six present the case studies of Kenya and Slovenia, whilst the seventh section brings them together in a comparative analysis, and section eight concludes the article.

*Relevance and Rationale*

Water is an essential resource for human survival; however, many countries in the world today face water-related issues, such as deterioration of water quality, water-related disasters, and water scarcity [5]. Population growth, economic development, and climate change are predicted to exacerbate and complicate these problems [5]. Water governance is also inherently complex due to being multisectoral and multilevel and subject to political negotiations among stakeholders with different interests [6,7]. The international human right to water and sanitation is part of a larger discussion on how water best can be governed to ensure sustainable management of water resources and access to water for all. Studying the mobilisation for the human right to water can thus give important insights into the nature of the contestation and discussions around water governance, including its priority on agendas, and into the public interest [5].

The case selection for this article is based on a most different systems design approach in which we looked for two cases with similar outcomes (constitutionalisation of the right to water in temporal proximity to the adoption of the international norm) but that differ on variables assumed to be relevant to explain this outcome, in this case s on factors shaping the conditions and opportunity structures of the local and international actors involved [8]. Our hypothesis was that relevant factors would be the material context related to water governance, the geo-political and normative context, and the structure of the decision-making process. We therefore looked for cases that differed in the levels and types of water governance challenges, among other due to differences in resource constraints and regional integration; where the geo-political context and the historical trajectories differed in ways potentially impacting the normative conditions related to social rights and state responsibility for distributive justice; and where the constitutionalisation processes differed in scope and openness of the decision-making structure. As illustrated in Table 1 below, the two countries we selected differ on these factors. They thus provide multifaceted insights into the processes of constitutionalising the right to water and mobilisation by both local activists and transnational norm entrepreneurs.

**Table 1.** Characteristics of cases (similarities and differences).

| Factors | Kenya | Slovenia | Similarities and Differences |
|---|---|---|---|
| Time of constitutionalisation process | 2002–2010 | 2013–2016 | Similar time period, proximity to international norm development. |
| Scope of the right compared to international norm (which includes sanitation) | "Every person has the right (…) to reasonable standards of sanitation; (…) to clean | "(R)ight to water for household use" indirectly including the right to sanitation | Similar in scope although sanitation is implicit in Slovenia. Both reflect scope of international right but with differences in wording. |

| | | | |
|---|---|---|---|
| | and safe water in adequate quantities" | | |
| Material context: water governance challenges | High levels of water scarcity, low government capacity | Increased prices due to privatisation, deteriorating water quality, high government capacity | Differ in water governance concerns and capacity. Water and sanitation challenges are larger in Kenya, while State capacity to address them is lower. |
| Geographical context | Regional influences from Africa, particularly South Africa | Regional influences from EU and Europe | Different regional context. Slovenia's integration in the EU provides a more comprehensive water governance framework. |
| Normative context: broader rights discourse | Socio-economic rights; right to life with dignity; health; food; housing; social security; education | Anti-privatisation, environmental rights; right to natural resources; sustainability | Differ in normative context: in Kenya, socio-economic right discourse is strong; in Slovenia, anti-privatisation, public ownership, and environmental rights are dominant. |
| Scope of constitutionalisation process | Part of new constitution | Constitutional amendment | Differ in attention to issue. Presumably more focus on water in Slovenia, where it was the sole focus of an amendment, than in Kenya, where the whole constitution was on the table. |
| Platform for decision-making/mobilisation | Constitutional review committee (and referendum) | Parliament | Differ in decision-making structure, with the process in Kenya presumably more open to bottom-up mobilisation compared to Slovenia with a parliamentary process. |

## 2. Theoretical Framework

A starting point for this article is the following question: How does the rights to secure, adequate access to water get constitutionalised, and how (if at all) does the process of global norm development influence the social and political construction of the right at the domestic level? We study the constitutionalisation of the right to water through a dual framework that simultaneously captures processes of norm diffusion from the international to the national level [3], the role of norm entrepreneurs [2] in this process, and domestic actors' agency, focusing on their opportunity structure [4]. Important for the latter are insights from the water governance literature clarifying how issues related to water and sanitation are and must be addressed in multiple sectors and at different levels of government and how stakeholders and activists engage at various points within the governance system. This section presents the key concepts and the overall theoretical lens.

### 2.1. Norm Diffusion and Norm Entrepreneurs

Finnemore and Sikkink presented a life-cycle framework for analysing the emergence and diffusion or cascading of international norms, distinguishing different stages in a norm's life, with distinct actors, motives, and mechanisms of change [3].

At the initial norm emergence stage, norm entrepreneurs and the work they do to promote a new norm is critical. Norm entrepreneurs are individuals, international organisations (IOs), and non-governmental organisations (NGOs) who want to change social or legal norms and who create or call attention to a new issue or norm by applying different strategies to "alert people to the existence of a shared complaint and [who] can suggest a

collective solution" [2,3]. Finnemore and Sikkink emphasised persuasion as a main diffusion mechanism at the norm emergence stage. Due to the generally low level of acceptance, it is difficult to make others adopt the norm without some form of pressure. The initial low acceptance or support for the new norm can also be caused by differences in how water is perceived [9], which have implications for how actors prioritise different aspects of water governance and the opportunities for improved governance.

When a critical mass of states (about one third of all states) has adopted the norm or a critical state (one without which the achievement of the substantive norm goal is compromised), Finnemore and Sikkink [3] argued that the process reaches a tipping point after which the norm is spread to states at a higher rate and without as much resistance. At this norm cascade stage, international socialisation is the main mechanism for norm diffusion or contagion. Socialisation comes in different forms, such as emulation and praise towards actors that advocate and follow the norm and ridicule towards the actors who do not follow the norm. These mechanisms can be performed by states, IOs, NGOs, and other network members. Institutionalising the norm in international rules or organisations can also constitute a critical juncture and take the norm across the threshold to the norm cascade stage by "clarifying what, exactly, the norm is and what constitutes violation and by spelling out specific procedures by which norm leaders coordinate disapproval and sanctions for norm breaking" [3].

Once the norm becomes widely accepted and internalised, behaviour conforming to the norm is habitualised, and there is a greater "taken-for-grantedness", the norm has reached the third stage of internalisation, where diffusion no longer relies on norm entrepreneurs.

According to Finnemore and Sikkink, not all norms reach the full life cycle. It is easier for norm entrepreneurs to speak "to aspects of belief systems or life worlds that transcend a specific cultural or political context" [3]. Hence, norms that involve bodily integrity and prevention of bodily harm or legal equality of opportunity are particularly effective transnationally and cross-culturally. Additionally, norms like the right to water, which concern the health and well-being of vulnerable and innocent people, resonate well with basic ideas of human dignity across borders and cultures.

Goodman and Jinks [10], expanding on the work of transnational norm diffusion, argued that processes of coercion and persuasion fail to account for a variety of ways in which social and legal norms diffuse. Their typology distinguishes between three types of mechanisms for social influence: coercion, persuasion, and acculturation. Coercion entails influencing behaviour by tipping the cost-benefit situation to reward conformity and punish nonconformity. It "does not necessarily involve any change in the target actor's underlying preferences … [but operates] by changing the cost-benefit calculations of the target state" [10]. Persuasion attempts to induce change in the belief or attitude of another person through transmission of a message and is a form of social learning [10,11]. Since norms never arise in a vacuum but emerge in a space with competing normative frameworks, an important persuasion strategy that norm entrepreneurs use is reframing the issue to make them resonate better with already accepted norms and values. Another persuasion mechanism is cuing, which targets audiences to engage in a process of "cognition, reflection, and argument" by introducing new information about the topic [10]. Acculturation is the process of conforming to beliefs and behaviour through socialisation with nearby and surrounding cultures, driven by both exogenous and endogenous pressures to assimilate. Actors are influenced by their social surroundings and change their behaviour and cognition accordingly. Acculturation thus operates through internal cognitive pressures (social-psychological costs of non-conformity; benefits of conformity; cognitive dissonance) as well as external social pressures (shaming, shunning, conferral of benefits through back-patting, and public approval) [10]. Any instance in which an actor or institution tries to influence the behaviour of another actor could include one or a combination of the features of all mechanisms.

When analysing the constitutionalisation of the right to water in Kenya and Slovenia, these theoretical perspectives on norm diffusion are used to understand the potential role of international norm entrepreneurs in the two processes and the significance of the different stages of international norm institutionalisation in which they take place. From the perspective of the domestic actors working to institutionalise the right, international norm development and norm entrepreneurs form part of their opportunity structure as potential resources and allies in the process or, in some cases, as back-seat drivers.

*2.2. Opportunity Structure*

When engaging in activities of norm development, such as constitutionalisation, the actors involved will, both consciously and subconsciously, consider their opportunity structure [4]. The opportunity structure is here understood as the sum of the internal and external resources and barriers that define the range of possible and opportune courses of action. This includes formal and institutional, financial, and historical factors that can influence and facilitate mobilisation of a cause through political channels, courts, or in social arenas [4,12,13]. Gloppen [4] distinguished between four different aspects of actors' opportunity structure: (1) The normative dimension refers to the resistance towards and support for the desired norm development in the society and the discursive resources available that might be mobilised to achieve the change, (2) the socio-economic dimension of the opportunity structure concerns the availability of the material resources required for different courses of action, and (3) the political opportunity structure refers to the openness of the political system to the actors and their concerns and the potential for achieving the desired norm change through the political process, while (4) the legal opportunity structure refers to the openness of the legal system and the availability of the resources required to advance the cause through legal mobilisation. Opportunity structures are not static or exogenous to the actors. Activists might shape or change them through their framing of the cause and through sequential "battles". The opportunity structure may also change as a consequence of external circumstances or actions taken by other actors in the field [4].

Especially interesting to our current analysis is the notion of normative opportunity structures and how different normative opportunity structures and the discursive strategies they allow can "determine the degree of visibility, resonance, and legitimacy" of a claim [4,14]. Similarly, in theories of norm diffusion, the shape and success of norm diffusion in a particular context vary according to the cultural match of the norm and the "receiving" context [15]. The more acceptance and proliferation there is of similar or supportive norms or claims within a context, the higher chances of success of the new norm; or, in other words, if a new norm resonates closely to existing normative values and claims, the higher chance of success [15]. Thus, the existing norms and values, the normative context of the society, is an important part of activists' opportunity structure. It will be influential in the choices that domestic activists and international norm entrepreneurs make, and on the impact the new norm will have.

Political opportunity structures will be shaped by the water governance structure in the given country. The spatial scale (how water governance is organised at different levels of government with different time frames and strategies) adds uncertainty and complexity [6] to water governance reforms and thus to the political opportunity structure of agents seeking change. With evidence suggesting an increase in the use of multi-level governance approaches in the water sector, there is a need to understand constitutionalisation processes also in light of the specific water governance context [9].

In the context of constitutionalising the right to water, we assume that in contexts where the ideas of human rights, socio-economic rights, and anti-privatisation movements are prevalent and widely accepted among relevant stakeholders or in the broader society, the right to water will resonate better and have a greater degree of acceptance. We also assume that the international norm development and the recognition of the human

right to water and sanitation in 2010 significantly strengthened the normative opportunity structure for actors mobilising for a right to water norm at national level.

### 3. Methods and Data

We use a mixed methods approach to identify the key actors in the "battle" for including the right to water and sanitation in Kenya and Slovenia, the type of discourse they relied on to argue for and against the right and particular framings, and which opportunity structures they utilised to achieve their cause. We use different types of data material and analytical approaches to extract the relevant information and details from the data. The primary data sources are documents from the parliamentary and constitutional processes and secondary literature. Interviews and written statements from activists engaged in the pro-water right group at international and domestic levels are used to contextualise the information.

For Kenya, we rely mainly on primary textual data material from the constitution-making process hosted by the Katiba Institute [16], which is a cooperation between the government and the national library in Kenya, established to promote the understanding and implementation of Kenya's Constitution. The archive contains documents from the constitution-making process [16]. Building on Loen's [1] study of the language in these documents, we identify discourses used during the constitution-making process. The Slovenian case is studied through parliamentary proceedings, secondary literature, and by written statements from activists and politicians engaged in the process of amending the constitution. The official website of the Slovenian National Assembly has a calendar of activities and meetings of the National Assembly, Državni Zbor [17], which allows us to identify when, where, and by whom the constitutional amendment was discussed.

To study the arguments and discourses around water and the right to water, we conducted text analysis of the collected documents. Moreover, we identified the actors involved by analysing documents, written statements, and interview data. Lastly, we conducted literature reviews and analysed interview transcriptions to identify and study the opportunity structures and mechanisms of diffusion in each country. This is a form of process tracing as we examine many intermediate steps in a process to make inferences about hypotheses on how that process took place and whether and how it generated the "outcome of interest" [18]. Process tracing is a fruitful way to study norm diffusion because of its ability to generate "empirical knowledge on decision-making processes, actors, and how their interactions produce the outcome of interest" [15,19,20].

However, for process tracing to be done well, it requires thorough knowledge about the case(s) itself, the theories and hypotheses that are being tested, and other, alternative explanations [21,22]. This is an exploratory study where there are still untested hypotheses and alternative explanations to be explored. The data base is not saturated since there are viewpoints and narratives that are not covered. Hence, we will not draw hard conclusions. Rather, we contribute insights into a larger field where these questions still are being asked. Furthermore, we believe that our research offers valuable knowledge on how constitutionalisation on the right to water and sanitation has been achieved. In a situation where 884 million people are without access to water and 2.6 billion with less than adequate sanitation and where implementation of the right to water is seen as a way to improve the situation for millions of people, these insights are highly relevant.

We start the presentation of our findings with an outline of the process towards developing a human right to water-norm at the international level and identify the key transnational norm entrepreneurs before looking at our two cases in more detail.

### 4. Transnational Norm Entrepreneurs and the Development of the Human Right to Water and Sanitation

Water, and to a lesser extent sanitation, has been on the international agenda for decades. We can trace water language back to the UN Water Conference in Mar del Plata in 1977 [23]. Although there has been a relatively broad consensus on the issue of water,

some actors have put in a greater effort to legally enshrine the right to water and sanitation in international documents. These are the actors we label norm entrepreneurs. Before we get to these actors, we will give a brief outline of the trajectory of the Human Right to Water and Sanitation.

### 4.1. The International Norm Development

Already in 1977, issues related to water management were addressed at the UN Water Conference. The conference emphasised that "efforts to improve the economic and social conditions of mankind, especially in the developing countries…will not be possible to ensure…unless specified and concerted action is taken to find solutions and to apply them at the national, regional and international levels" [23]. It is also stated that "All peoples, whatever their stage of development and their social and economic conditions, have the right to have access to drinking water in quantities and of a quality to their basic needs" [23]. The International Convention on Economic, Social, and Cultural Rights (ICESCR) from 1966 did not include the right to water and sanitation [24], but the right to water was later inferred from the right to an adequate standard of living, notably in General Comment No. 15 by the Committee on Economic Social and Cultural Rights [25].

That multiple international water and sanitation decades have been proclaimed in recent history also illustrates the international community's attention towards and consensus on the importance of water related issues. These include the International Decade for Clean Drinking Water (1981–1990), the International Decade for Action "Water for Life" (2005–2015), and the International Decade for Action on Water for Sustainable Development (2018–2028) [23,26,27].

Following the adoption of the ICESCR in 1966, several international conventions contained an explicit right to water. In 1979, the Convention on the Elimination of All Forms of Discrimination Against Women declared that states are responsible for ensuring women the right "to enjoy adequate living conditions, particularly in relation to housing, sanitation, electricity and water supply, transport, and communications" [28]. The Convention on the Rights of the Child of 1990 states that children have the right to the highest attainable standard of health, which includes the "provision of adequate nutritious foods and clean drinking water" [29]. Additionally, the 2006 Convention on the Rights of Persons with Disabilities maintains the need for clean water services [30].

However, these treaties do not regard water as an independent right but rather as an essential component of other rights, most notably as the right to health and the right to an adequate standard of living. Not until the Committee of Economic, Social, and Cultural Rights issued General Comment No. 15 in 2002 was the right to water explicitly mentioned as a self-standing, independent right [25], and it was finally recognised as an independent human right in resolution 64/292 of 2010 [31].

### 4.2. Central Actors at the International Level

Madeline Baer argues that the path towards acceptance of the human right to water and sanitation differs from other processes of new rights emergence because much of the work happened outside the human rights regime and without the active involvement of traditional rights gatekeepers [32]. However, some traditional rights gatekeepers have been involved in the process, including the Human Rights Council; The Committee on Economic, Social, and Cultural Rights; and the United Nations General Assembly. As noted, the Committee issued General Comment No. 15 in 2002 [25], where the right to water is derived from the ICESCR's Art. 11, which states that "[t]he States Parties to the present Covenant recognize the right of everyone to an adequate standard of living for himself and his family, including adequate food, clothing, and housing, and to the continuous improvement of living conditions". Based on this, General Comment No. 15 states:

> The use of the word "including" indicates that this catalogue of rights was not intended to be exhaustive. The right to water clearly falls within the category of

guarantees essential for securing an adequate standard of living, particularly since it is one of the most fundamental conditions for survival. [25]

The right to sanitation was not recognised as an individual right in General Comment no 15, but in 2006, the Human Rights Council (HRC) gave the Office of the High Commissioner for Human Rights the mandate to conduct "a detailed study on the scope and content of the relevant human rights obligations related to equitable access to safe drinking water and sanitation under international human rights instruments" [33,34]. The HRC planned a three-folded mobilisation process for the right to water and sanitation, of which the study was the first step. The next step was to appoint an independent expert that would develop a dialogue with stakeholders, work on best practices related to access to safe drinking water and sanitation and make recommendations to help realise Millennium Development Goal No. 7 [35]. Lastly, they would advocate for an independent and explicit recognition of the right to water and sanitation [36].

In 2008, Catarina de Albuquerque was appointed The Independent Expert on the Issue of Human Rights Obligations Related to Access to Safe Drinking Water and Sanitation (later, the Special Rapporteur on the Human Rights to Safe Drinking Water and Sanitation) [35]. The Special Rapporteur mobilises by conducting country visits, preparing thematic research, and cooperating with practitioners, stakeholders, and government to raise awareness of water and sanitation issues and mobilise support for recognizing these as human rights concerns [37]. During her first years in the mandate, de Albuquerque worked hard to advocate for the need to have an explicit right to water and build consensus around this idea [38].

Both the HRC and the Special Rapporteur placed focus on building political consensus around the right to water and sanitation. Informal meetings with NGOs and civil society organisations were held, and consultative meetings allowed states with different objections or worries to express them and come up with solutions [38]. In 2010, the main resolution on the Human Right to Drinking Water and Sanitation was adopted in the United Nations General Assembly, with 122 countries voting in favour.

Baer [32] adopted the terms champions and challengers when discussing the fight for the right to water and sanitation, and Bolivia is certainly among the right to water champions and a prominent actor in the anti-privatisation movement. Bolivia was a main architect of the draft resolution, which was co-authored and sponsored by several additional countries, including Uruguay, Ecuador, Nicaragua, Spain, and Germany [33,34]. The Independent Expert also contributed in important ways by ensuring that sanitation was ultimately included in the resolution [38].

This overview illustrates that there are many important international and transnational actors working towards establishing a right to water at the international level. Some of them were also connected to parallel national processes of constitutionalising the right to water, for example, in Kenya. The General Comment No. 15 was written by people with close ties to local and transnational organisations that were well-established in Kenya in the early 2000s [38]. We therefore expect that several of the people associated with the United Nations agencies, particularly the Committee on Social, Economic, and Cultural Rights, were prominent in the Kenyan context and part of the norm environment. Moreover, due to South Africa's progressive constitution in terms of socio-economic rights, we see South Africa as a critical state both in the African region and worldwide [39]. Due to the proximity of South Africa and Kenya, we expect there to be a strong influence. In Slovenia, we believe that the visit from the Independent Expert have contributed to some norm diffusion. We also expect some influence from the actors who were most vocal against privatisation (Bolivia, Project Blue Planet). Additionally, we believe that there will be involvement from the EU to comply with the Copenhagen criteria on competition and open-market policies [40]. In the case studies presented below, we illustrate how some of these actors played a part in the constitutionalisation process, while others were absent.

## 5. Kenya

In this section, we present our findings from the Kenya case study. It suggests that including the right to water in the constitution responds to a colonial past and the country's subsequent economic and political history and reflects a determination to reduce poverty and inequality through development.

Kenya's new constitution was adopted in 2010, and article 43 on Economic and Social rights explicitly states that all Kenyans have the right "to reasonable standards of sanitation" and "to clean and safe water in adequate quantities". This was the outcome of more than a decade of constitution-making following the adoption of the Constitution of Kenyan Review Act in 1998 and the swearing in of the Constitution of Kenya Review Commission two years later [41]. The Review Commission was responsible for providing civic education, seeking the issues and views of the people, and preparing a draft constitution for a National Constitutional Conference (NCC) [42]. Political turbulence and changes in government put a halt to the constitutional review process, but in 2009, a Harmonised draft was finalised, and on 4 August 2010, it was accepted in a referendum.

The right to water and sanitation was included both in the early drafts and in the final constitution but changed during the decade-long process. The first draft, adopted at the National Constitutional Conference in 2004, included individual articles for the right to water and the right to sanitation. In a 2005 draft, both articles were removed, while the Harmonised Draft, which was presented by the Committee of Experts in 2009, reintroduced a free-standing right to water, while sanitation had now become a part of the right to housing. The President's party again tried to remove the right to sanitation, but the constitution adopted in 2010 brought back sanitation as part of the right to housing [1].

### 5.1. Key Actors

According to the official documents, the main actors in the constitution-making process in Kenya are official actors. These include the Review Committee and its sub-committees, The Committee of Experts (hereafter the Expert Committee), politicians, parliamentarians, and members of the government and the administration [1]. In addition, input and suggestions came from citizens, societal groups, NGOs, civil society organisations, external actors, and experts. External actors is a broad term, including international human rights activists, constitution experts, UN employees, and foreign academics (for a detailed overview, see Loen [1]). References to the Review and Expert Committees are present in a majority of the documents where water is an issue, while NGOs, civil society organisations, and individual persons or a combination are present in a quarter of these documents [1]. The external actors are not present in many of the documents (low frequency), but due to the expert status of these individuals and organisations, we believe that it is important to include them in the analysis. These include states in relatively close proximity with similar rights discourse (South Africa, Tanzania, Uganda, and Ethiopia), civil society organisations and local NGOs, international non-governmental organisations (Amnesty International, Wash United, Centre on Housing Rights and Evictions), foreign governments and agencies (German Society for International Cooperation/Deutsche Gesellschaft für Internationale Zusammenarbeit (GIZ)), and international governmental organisation (United Nations bodies, WTO, IMF).

### 5.2. Political Opportunity Structures

The political system and culture of Kenya facilitated a participatory and open constitution-making process that provided actors with opportunities to influence the contents of the constitution. This should be understood against the background of Kenyan history. Following British colonial rule (1920–1963), Jomo Kenyatta became the first Kenyan president. He introduced an authoritarian regime, which was maintained by Danial Arap Moi until multiparty elections and other democratic reforms were introduced in 1991. Due to

weak opposition and an uneven playing field, Moi and his KANU party won the following two elections and remained in power until conceding to Kibaki in 2002. With the reintroduction of multi-party elections in 1991, a culture of resistance and human rights emerged that triggered and infused the constitution-making process [43].

The official documents, several of which are verbatim reports from public hearings and meetings held over a period of two years around the country, suggest that the pro-water right actors faced a relatively open political opportunity structure. Public hearings and meetings convened by political leaders and the Review Committee were used to create dialogue and public participation. The meetings were platforms for the public to speak about what they wanted to have in the constitution. The meetings always consisted of representatives from the Review Committee, but the meetings had different topics, and some were hosted especially for certain groups, such as women's representatives, children, and religious groups. This was meant to increase public participation but also to give different groups an arena to lift their considerations and inputs. The Review and Expert Committees also collected written, recorded, or otherwise conveyed information from citizens regarding their perspectives, views, and interests. External actors were invited to participate in public meetings and topical seminars and to join the Committee of Experts, and the media ensured that the Review and Expert Committees could convey information and education to the public. The political system was seemingly responsive to the public grievances, and there was little overt opposition towards constitutionalising the right to water. At the same time, some government partisans repeatedly removed the articles on water and sanitation from the constitutional draft, and it is reasonable to assume that this would have prevailed had the previous government and president stayed on in power.

### 5.3. Normative Opportunity Structures

The key normative framework that pro-water-and-sanitation-rights actors in Kenya build on is development, which is strongly related to Kenya's high levels of poverty and inequality. The authoritarian regime that came into power after independence from Great Britain in 1963 maintained the liberal economic system introduced by the colonial rulers [41–45]. However, the period after independence also brought low state capacity, inequality, and poverty. This became central to the struggle for democratic and economic reforms during the 1980s.

We started the search in the 271 documents from the constitution-making process. After excluding the irrelevant documents, we ended up with 84 documents containing references to water and 48 documents referencing sanitation [1]. The documents span ten years, from 26 March 2001 to 5 December 2011. Mentions of water and sanitation in the documents have been coded into categories of types of discourse, actors, and other relevant categories (the coding scheme is provided Table A1 in Appendix A). Several actors participated in shaping the constitutional draft. We argue that the international norm development that had been ongoing for many years and accelerated in the decade leading up to the adoption of the constitution greatly shaped the normative opportunity structures for water and sanitation rights activists in Kenya and that certain regional and national factors also have been influential for the acceptance of and adherence to a right to water norm.

The document analysis shows that the most frequent frame for both water and sanitation is right (63 and 27 documents, respectively; this includes references to water as a resource that citizens have the "right" to access, as well as the "right" to water, and water as a "human right"). The second highest frequency is groups and provision for water and sanitation, respectively. *Groups* refers mainly to marginalised groups, minorities, informal settlers, and women, who are particularly vulnerable to water scarcity, whilst *provision* encompasses references to the responsibility of the state to provide water. The rights to water and sanitation are also mentioned with reference to the following categories: access (to water or lack thereof), natural resources and environment (protection of natural water

sources, sustainable usage of water resources), low-income groups (need for affordable water and sanitation services for low-income groups), health (the importance of clean water and sanitation for public and personal health), custody (the right to standards of sanitation for persons held in custody), and housing (right to sanitation as part of right to housing). There is a great deal of inequality in Kenya, and socio-economic and geographical cleavages overlap. This is reflected in the documents, which frequently mention areas where the level of poverty and water scarcity is much higher.

In terms of actor categories, state actors most frequently refer to water and sanitation as rights. Provision is also often mentioned as the responsibility of the state, and they also discuss water as natural resource and a part of the environment that must be protected. The provision and groups references to water are similar in the context of sanitation. However, sanitation is also brought up in relation to groups of people living in deplorable conditions in Nairobi.

The public and civil society groups often refer to water and sanitation in the context of rights, groups, access, and low-income. They are concerned with the lack of clean and affordable water for people in the poorer districts and regions, slum-dwellers, and informal settlers. They worry about women who must walk many kilometres every day to collect and carry water in kegs and *mitungis*, and they want the government to ensure access to clean piped water to all citizens.

Civil society and citizens clearly expected human rights to be anchored in law through constitutionalisation, and as is clear from the discussion above, the right to water and sanitation received attention and support during the constitution-making process. Members of the public were highly concerned with marginalised groups and the lack of equality among sociodemographic groups, such as people in the northern districts of the Eastern, North-Eastern and Rift Valley Provinces. The people who live there "are deprived of the same chances for education, of access to water, and of security in comparison with those in most other parts of the country". Similarly, poor and marginalised groups "are deprived of access to basic needs especially education, medical care, housing, transport, sanitation", and "lack access to basic amenities, such as water, food, and shelter". It is also evident from the Review Committee's final report that the people:

> … expected that the new Constitution would take into account the needs and aspirations of the disadvantaged and marginalised members of society. In many respects, they expected the new Constitution to solve a myriad of socio-economic problems and create a drastic improvement in their livelihood, especially alleviate poverty, eradicate corruption, create employment opportunities, and provide adequate food, shelter, health, education, water, and land for every Kenyan (sic). [16]

We assumed that South Africa's 1996 constitution would be a significant regional influence, and our analysis does show signs of South African influence. Based on the findings we present below, it is likely to assume that some form of norm diffusion took place through mechanisms of persuasion, acculturation, and coercion. The prominent South African Human Rights advocate Geoff Budlender was invited to provide the Review Committee and Standing Committee on Human Rights with knowledge and experience from the South African constitution. He offered detailed insights into how the South African constitution provides routines for promoting and securing human rights and for allocating resources for progressive realisation of rights and how affirmative action is used to protect marginalised and vulnerable groups. This is a way of inducing the listeners to the belief that safeguarding human rights should be done by securing them in the constitution, as was done in South Africa. As demonstrated in the excerpts below, Budlender emphasised how South Africa found inspiration for their constitution in international documents. By framing this as an international standard, we might even claim that it amounts to a form of coercion by appealing to conformity and cuing Kenyans to see this as the appropriate behaviour:

The international community has long realized that for our inherent dignity and right to life to be respected, the material conditions of our lives must be such that it is possible. […] That is recognized from long ago in 1948 by the Universal Declaration of Human Rights, which deals very explicitly with the conditions of life, deals very explicitly with the need for matters such as inadequate standards of living including food, clothing, housing, medical care, and social services […]. In South Africa, what we did was we followed the structure of the international covenant on economic, social, and cultural rights. We said we would have a general statement of the rights followed by the description of the duties. You have got [… a] copy of our bill of rights, and [if] you turn later to Section 26 of that, you will see the housing right, which explains how we have tried to deal with it. Let me turn to that. Section 26 I of our bill of rights of our Constitution contains a general statement of the right. Everyone has the right to have access to adequate housing; it is a fundamental right, which everyone has to have access to […], and it is the general statement of the right. [16]

Other important external actors are donor and development agencies, such as the German development agency GTZ (which became part of GIZ in 2011). The project "Realising Human Rights in Development Cooperation" aimed to improve and develop the water sector in Kenya through a water sector reform [46]. As part of the reform, Kenya adopted the 2002 Water Act, which treats water as an economic good [47]. However, the assistance from GTZ was not only about commodification of water; the organisation also aimed to implement the reform through a human rights-based approach [48]. Therefore, whilst funding new water policies and encouraging commodification of water services, they also advocated for the human right to water. This was not seen as contradictory; rather, GTZ argued that economic reform would enhance the provision of and access to water for all citizens [48]. After surveying citizens, they found that there was a lack of knowledge and understanding of the costs and resources it takes to run water and sanitation services but that consumers see access to water supply as a right and that there is a "high consumer sense of responsibility for payment for water consumption" [49]. It was thus a goal to inform the public on the ins and outs of water management and services and to raise awareness on water shortage in the country, and in 2004, the GTZ adopted a communication strategy for the Water Act that would focus on the:

… use of community-based social, religious, and civic/ political organisations, individuals, and networks in Kenyan society as channels and influencers to communicate with people "face-to-face". Examples would be speaking through women's groups, barazas, and church groups, etc. A radio entertainment-educational serial drama linked to community level activities is also recommended as a central activity for this phase. [49]

GTZ thus participated in water right advocacy both through mechanisms of coercion and persuasion.

## 6. Slovenia

In 2016, Slovenia amended its constitution to include Article 70a, which states that everyone has the right to drinking water and that water is a public good subject to the authority of the state [50]. Slovenia is one of only three European states that have constitutionalised the right to water, along with Iceland and Hungary [50]. In this section, we outline the process that led to the constitutional amendment in 2016 and trace the interaction of the central domestic actors with international norm entrepreneurs. We find that the right-to-water campaign strongly reflects an anti-privatisation discourse, the political culture, and history of state ownership and nationalism as well as Slovenia's European Union membership.

In terms of water resources, Slovenia is one of the richest countries in Europe. Located in the middle of Europe, with mountainous topography, its access to the Adriatic

and Mediterranean seas, and its many underground rivers, gorges, and caves, Slovenia has access to great amounts of groundwater and surface water [51]. While this is a great foundation for providing accessible, affordable quality drinking water to all its citizens, the large surplus of relatively cheap water is also an attractive commodity for foreign companies [51]. There are several examples of foreign companies buying local breweries and local water suppliers or getting concessions for water use, causing an increase in drinking water costs and deterioration of water quality. The Dutch brewing company, Heineken, for example, took over two local breweries and bought a local water supply in Laško, thus causing a 30 per cent increase in drinking water costs for the city's inhabitants [51]. Another company acquiring a concession for a water purification plant caused a deterioration of water quality [51]. These developments led to the emergence of advocacy for constitutionalising the right to water starting in 2013.

In 2014, fear of a proposed EU directive on the awarding of concession contracts gave fuel to the advocacy, where civil society activists joined forces with parliamentarians [51]. If an EU directive is adopted, EU Member States must adopt their national legislation to EU law. This particular directive would, among other things, regulate privatisation of Member States' water resources. There were strong concerns about this directive among several Member States because of the involvement of entities in sectors that would benefit from its adoption, particularly private companies in need of water resources [51,52].

### 6.1. Key Actors

There are two main groups of actors in the Slovenian case. Firstly, several political leaders and parliamentarians were engaged in the efforts to amend the constitution. Secondly, there was an incredible mobilisation and support from the public. In the aftermath of the proposed EU directive, Slovenian parliamentarians proposed to include the right to water in the constitution. They wanted to protect the water resources and Slovenians' access to drinking water from future privatisation legislation. There was broad consensus among parliamentarians the necessity of a constitutional amendment, and the first proposal was put forward by a coalition of 35 legislators [52].

The public mobilisation also played an important role, sending strong messages that water is a public good, water resources should not be privatised, and water supply for the population cannot be carried out as a profit-oriented activity. An interesting point of study is the interplay between political parties and the public mobilisation. A prominent person in the right-to-water mobilisation in Slovenia stated that "political parties are encouraged by the people to include the protection of water resources and the protection of nature in general in their political agendas" [53].

Very important in the Slovenian context was a civilian initiative consisting of lawyers, journalists, programmers, filmmakers, national and European politicians, and volunteers contributing skills, knowledge, and determination [54]. The lawyers provided advice on the legal obstacles and opportunities for constitutionalising the right to water, whilst the filmmaker created "short videos with celebrities, which had a big impact to the people" [54]. The initiative used several strategies to raise awareness and encourage debate the issue, from social media to lectures and events. The analysis we present below suggests that the civilian initiative spread awareness and generated support for the right to water through mechanisms of socialisation and acculturation.

### 6.2. Political and Socio-Economic Opportunity Structures

Slovenia's political and geopolitical history is unique, and especially the socialist Yugoslavian legacy has been important both for Slovenia's position and opportunities for integration into the European Union and for democratisation and modernisation opportunities [55]. Starting in the late 1980s, Slovenia introduced several liberalising reforms, including multi-party elections, toleration, and eventually promotion of pluralism and

diversity [55,56]. Slovenia also started an enduring process of European integration, economic transformation, and privatisation motivated by a strong democratic, economically liberal, pro-Western orientation among both citizens and elites [55,56].

Slovenia emerged from the Yugoslavian federation with a low debt burden compared to other Yugoslavian states due to its successful negotiations with IMF and international lenders [57], and their initial conditions for development were more favourable than for most other Central and Eastern European countries. They also continued their left-wing government tradition, taking a more modest approach towards market liberalisation and privatisation also in their accession negotiations with the EU (favouring domestic owners, shares to state-controlled funds, employees, and internal buyouts). The pressure from the EU to privatise businesses and shares caused strong domestic resistance, which was "embedded in a domestic consensus surrounding the advantages and ultimate success of Slovenia's less radical transition path" [57]. The ambition to privatise the economy created political turmoil, but eventually, a privatisation plan and legislation allowing for sale and free distribution of state enterprise stocks was drafted, and in November 1992, it was adopted by the parliament [58]. The transition to a market economy is an important backdrop of the process of constitutionalising the right to water in 2016.

In the early 2000s, when a Slovenian brewery was faced with pressure to sell shares to a foreign company, Slovenians united in their opposition towards "the perceived threat of a foreign takeover of a beloved national brewery" [57]. It also prompted a more general public debate on foreign versus domestic ownership. Those who were sceptical towards foreign ownerships argued that the only motivation for the foreign actors is profit and that they specifically seek out the less developed European countries to take advantage of them, whilst those in favour argued that it would improve economic integration, and they also referred to the second Copenhagen criterion on "a functioning market economy and the capacity to cope with competition and market forces in the EU" [57].

The increased privatisation coupled with a fear of losing autonomy towards the EU on concession contracts gave fuel to the advocacy for constitutionalising the right to water that started in 2013 [53]. In 2015, the Civil Initiative for Slovenia and Freedom was formed. This is an informal connection of people "with different skills and of various professions, ages, ideological, and religious beliefs" [54] who took part in a campaign to promote the constitutionalisation of the right to drinking water, which was their main and only issue.

One particularly important member of the civilian initiative is Brane Gulobović, a former parliamentarian with knowledge on how to go from mobilising the issue to actually implementing policies and new laws. He became a pivotal actor in the initiative but also in the formal political system, as he had connections and ties to sitting parliamentarians.

During his term in parliament, he initiated the process towards amending the constitution. In August 2013, the Parliamentary Committee on Agriculture, Forestry, Food, and the Environment organised a public hearing on how to best ensure and protect the right to drinking water [17], and in March 2014, Gulobović and other members of parliament submitted a proposal to initiate the procedure of amending the constitution [52,59]. The process was disrupted by early elections, in which Gulobović was not re-elected, but in June 2014, the Constitutional Commission organised a public event on drinking water, where the proposed Brane Gulobović's draft amendment was discussed [59]. Two years later, the Civilian Initiative for Slovenia and Freedom met with the President of the National Assembly and handed over 45,000 signatures supporting inclusion of the right to drinking water in the constitution [60].

In July 2016, a constitutional amendment was proposed to the Constitutional Commission to enshrine in the constitution the provision that everyone has the right to safe drinking water. On 3 November, the Commission approved the proposal. The President of the National Assembly emphasized that he would do everything possible to complete the process of signing the Constitution as soon as possible, and already, in mid-November, the National Assembly discussed amending Article 70A of the Constitution of the

Republic of Slovenia, adding the Right to Drinking Water. The amendment was adopted by 64 votes in favour and 0 against [61]. A week later, the National Assembly met for an extraordinary session to promulgate the constitutional amendment [60].

There were few opponents to constitutionalising the right to water in Slovenia. The politicians generally supported the amendment from the start, and any reluctance in the political leadership was removed by the civilian initiative and the signatures of 45,000 citizens.

*6.3. Normative Opportunity Structures*

The normative opportunity structure of the water rights activists should be understood against the backdrop of Slovenia's political history and ideology. The literature that exists on Slovenia's slow and modest transition strategy shows that despite the wish to integrate into the European Union, the country held on to a national patriotism and socialist traditions [57]. Anti-privatisation positions have constitutional support. Article 2 of the constitution reads: "Slovenia is a state governed by the rule of law and a social state", and the constitution has strong corporatist features and emphasis on workers' rights [57]. The public's perception and understanding of terms such as public good is also highly contingent on the country's socialist past. Citizens have strong feelings about social justice, equality, and access to goods for everyone [62]. Natural resources, such as water, wild-growing foods, air and forests, peace, infrastructure (municipal properties, roads, paths, wells, ponds, monuments, and viewpoints), and public services, are all viewed as public goods or common goods by local Slovenes [62]. This sentiment was shared by the civilian initiative:

> *Our main goal was to be clearly written into the Constitution that water and water land is a natural public good, over which no-one can acquire ownership rights; that everyone has the right to drinking water; that the water supply of the population cannot be owned by private companies in any legal-formal way, and that the provision of the water supply to the public is a service which should not generate profit and that the water supply of the population has the absolute precedence over economic exploitation in the case of the water crisis or drought or other crises, and that the water resources be managed sustainably, with thoughts on our posterity.* [53]

The first appearance of water as a topic in parliamentary documents that we have identified is from August 2013, shortly after Brane Golubović entered parliament. Most of the parliamentary debates on this topic relate to anti-privatisation and efforts to prevent profit-oriented water supply [53]. Because of the water abundance in Slovenia and the high quality of the groundwater, there is little focus on contemporary problems of water distribution and access, but rather, the focus is on protecting future water provision and ensuring access of high-quality and affordable drinking water to future generations of Slovenes.

## 7. Discussion

Both in Kenya and Slovenia, the processes of constitutionalising the right to water involved a vibrant and participatory civil society and emphasised water as essential for current and future generations' health and quality of life. However, there are also many differences, including the sites of reform, the nature of the actors involved, and their ties to international pro-water right actors.

*7.1. Political Structures and Contexts*

In Kenya, advocacy for constitutionalising the norm mainly took place before the constitutional review committee, a body constituted for the purpose, whereas in Slovenia, it took place in and was initiated by Parliament. In Kenya, parties were actively involved in the constitution-making process, but the final decision was made in a referendum (bottom-up). In Slovenia, the decision was made in Parliament (top-down), but the process

was characterised by public participation and a broad public debate on the topic. In Parliament, the amendment was supported by broad coalition and accepted by consensus.

Whilst Kenya's 2010 constitution was developed alongside the international process of recognising the human right to water and sanitation, Slovenia's adoption of a constitutional right to water in 2016 took place in a somewhat different international context with a more firmly institutionalised international human right to water but also less international focus on the issue. While not a straight-forward process, enshrining the right to water in the Slovenian constitution was a significantly quicker process than in Kenya.

Finally, the regional context differs. Slovenia, as a member of the European Union, is integrated in an institutional context with stronger implications for domestic norm development. As we have seen, the constitutional amendment came as a counter-initiative to a suggested EU directive allowing for more privatisation. Kenya's regional context, while more loosely integrated, was one in which other countries already had constitutionalised the right to water (South Africa, Gambia, Uganda, and Ethiopia between 1994–1996) or were in the process of doing so (Niger in 2010, Somalia in 2012, and Eswatini in 2013) and where the South African influence in particular seems to have been significant.

### 7.2. Framing of the Right to Water

In Kenya, actors actively use rights language and primarily economic and social rights in their discussions on water and sanitation access. There is a strong emphasis on development, lifting people out of poverty, and ensuring access to health services, school, work, and basic needs, such as food and water, for all Kenyans. In the documents, we find that some state actors rely heavily on human rights discourse, with a main focus on the right to housing and the right to health. This illustrates how framing a new norm in a similar manner to already existing norms and rights increases chances of the new norm being understood, accepted, and secured by the actors within the state [3,10,14]. Additionally, in Slovenia, pro-water right activists used an existing framework of rights and values to generate support and acceptance for the right to water norm among citizens and politicians. The discursive opportunity structure favoured a contestation of the privatisation trends of the past decades and opposition of attempts from the EU to force privatisation in water-resource dependent sectors. Unlike in Kenya, where privatisation was argued by some actors as a route to ensuring universal access to and quality of water, it was broadly perceived as a threat in Slovenia.

The citizens and organisations who participated in the discussions in Kenya used a similar language as the state actors but stressed the importance of the state's responsibility for providing for socio-economic rights, including water. While there were prominent human rights advocates among the state actors, evidence suggests resistance from within the government. In some documents, the right is referred to as the right to access water and not merely the right to water. This has been suggested as a strategy for subliminally transferring their obligations to private actors [32]. This suggests that while the state was willing to accept a limited right to water for their citizens, they were reluctant to commit to ensuring all citizens access to high-quality drinking water. This is not a unique challenge. It is easy to sign off on a document giving citizens de jure rights but difficult to ensure de facto realisation. The Sustainable Development Goals created specific target goals as an attempt to administer this challenge [63].

### 7.3. Links to International Actors and Discourses

Analysing the processes of constitutionalising the right to water in Kenya and Slovenia, we find little evidence supporting a strong effect of the international recognition of the Human Right to Water and Sanitation in UN Resolution 64/292. Especially in Slovenia, where the right to water was already rooted into the discourse of anti-privatisation and a desire to preserve natural resources in national ownership, we see that the advocates had little need for a new normative framework. As noted by of the most prevalent activists:

There were no special contacts between our civic initiative and other NGOs across Europe, nor did we follow the example of some other countries that constitutionalised the right to water. [53]

Nevertheless, there are influences from external actors in both countries. While not central transnational norm entrepreneurs in the international mobilisation for a human right to water, they are of importance in the two cases. In Kenya, participants in the discussion repeatedly referenced the UN Convention on Economic, Social and Cultural Rights and, more specifically, rights to housing, life, and health in discussions on the right to water and sanitation. The analysis also suggests that citizens, organisations, and state officials concerned with the lack of adequate services and facilities framed the right to water as a state obligation. This finding suggests a stronger case of emulation international conventions and mechanisms in Kenya. The South African constitution also constituted a reference document from which Kenya found inspiration to include a strong bill of rights and mechanisms for monitoring and protecting these rights in the constitution. Last but not least, international donors and GTZ in particular influenced the discourses on water, both in terms of privatisation and rights-based approaches. In Slovenia, the scope of a domestically existing norm on state ownership, national patriotism, welfare, and protection and conservation of natural resources was widened based on their existing political culture and norms. However, the pro-water right movement in Slovenia grew in response to the EU directive on concession contracts, which was lobbied for by private companies looking for ways to increase profit [51].

## 8. Conclusions

As we have shown, to understand the constitutionalisation of the right to water in the two cases, the central actors' normative opportunity structures are key. The different ways of framing the right to water in the two cases demonstrate that alternative framings may be equally successful in creating support and acceptance for the norm. The domestic process does not necessarily have to rely on the United Nations-adopted norm on the human right to water and sanitation. The case studies show that to frame the right to water in relation to other human rights, development, environmental conservation or anti-privatisation can also be effective, if it resonates with the normative context.

There is clear evidence that favourable domestic political opportunity structures were important in both cases, illustrated by the pro-water right activists' usage of open political channels and many options for influence. Additionally, we find that domestic pro-water right actors use mechanisms of socialisation and acculturation to proliferate relevant information about the norm and to generate support for it, whilst international actors to a lesser extent influenced norm diffusion. When they have done so, such as GIZ and South African constitutional experts in the Kenyan case, they have used mechanisms of coercion and persuasion. In Slovenia, the main external influence seems to be the reaction sparked by the coercive influence of the EU directive.

The findings are in line with the expectations that normative opportunity structures are important. However, in contrast to our expectations, we find that the potential structures for advocacy opened by the international norm development and the recognition of the human right to water and sanitation in 2010 rarely were utilised by actors in Kenya and Slovenia. We believe that more research is needed on how international right norms are diffused, how domestic opportunity structures influence the acceptability of the norm at national level, and how actors can manipulate and change these structures. There is also need for more research on the effects of constitutionalisation of the right to water on citizens' possibility to enjoy their right.

**Author Contributions:** Conceptualization, M.L. and S.G.; methodology, M.L. and S.G.; software, M.L.; validation, M.L. and S.G.; formal analysis, M.L.; investigation, M.L.; resources, M.L.; data curation, M.L.; writing—original draft preparation, M.L. and S.G.; writing—review and editing, M.L.

and S.G.; visualization, M.L. and S.G. All authors have read and agreed to the published version of the manuscript.

**Funding:** Research funded by the Norwegian Research Council. "Elevating Water Rights to Human Rights: Has it strengthened marginalized peoples' claim for water? (Forskerprosjekt—FRI-HUMSAM project number 263096). P.I. Bruce M. Wilson.

**Institutional Review Board Statement:** Since the archival data materials were publically available and the interviewees were experts interviewed in their public capacity, and also were anonymised, the Review Board did not consider ethical approval required.

**Informed Consent Statement:** Informed consent has been obtained from the interviewees to publish this paper.

**Data Availability Statement:** The following data presented and analysed in this study is available: Constitutions of Kenya and Slovenia can be accessed from: Comparative Constitution Project. Constitute project. https://www.constituteproject.org/search?lang=en. (accessed on 28. July 2020). Parliamentary documents from the Slovenian Parliament can be accessed from: Državni Zbor. Kronologija VI. Mandata. https://www.dz-rs.si/wps/portal/Home/is/kronologija/!ut/p/z1/04_Sj9CPykssy0xPLM-nMz0vMAfIjo8zinfyCTD293Q0N3L2cTAwCjf19nYL-MgwyDPQz0w8EKvCy9Hb3ACoyCTA0CXYycfIMNjA2CjQz0o4jRb4ADOBKpH4-CKPzGF-SGhoY6KioCAIQMZuY!/dz/d5/L2dBISEvZ0FBIS9nQSEh/ (accessed on 11 October 2021). Documents from the Constitution of Kenya Review Committee were downloaded from: Katiba Institute. Katiba Digital Resource Database. http://www.katibainstitute.org/Archives/ (accessed on 18. November 2019). The Katiba Archive website has since been removed. Documents that were downloaded from the website and used in the analysis can be obtained from the corresponding author, along with the code syntax used to download the documents from the website through the RStudio software.

**Acknowledgments:** We are grateful to our colleagues at the CMI-UiB Centre on Law & Social Transformation; the Chr. Michelsen Institute, and the University of Bergen for comments, suggestions, and great conversations about this topic -- and particularly to the team in the research project "Elevating water rights to human rights: Has it strengthened marginalized peoples' claim for water?" (Researcher project—FRIHUMSAM project number 263096).

**Conflicts of Interest:** The authors declare no conflict of interest.

## Appendix A

**Table A1.** Coding categories.

| Variable | | Coding |
|---|---|---|
| Type of document | | Constitution (drafts and old constitution included) |
| | | Paper |
| | | Report |
| | | Working document |
| Present actors or author of document | State actors | Review Committee |
| | | Expert Committee |
| | | Special and topical committees |
| | | Politicians and parliamentarians |
| | NGOs and CSOs | NGOs or civil society organisations |
| | The People | Private persons, the people |
| | | Representatives of groups in society |
| | Professionals | Scholars, academics, professionals |
| | NA | Not applicable |
| Water | | Yes |
| | | No |

| Sanitation | Yes |
| | No |
| --- | --- |
| | Right / human right |
| | Minorities, marginalised groups (women, children, pastoralists, informal settlers) |
| | Persons held in custody |
| | Responsibility for provision |
| Water and sanitation—categories | Low-income groups |
| | Custody |
| | Natural resources and environment |
| | Inequality (geographical, social, in access) |
| | Health |
| | Provision |
| NGOs or civil society actors | Names of the NGOs and civil society actors |

Sources: Documents collected from Katiba Institute n.d. [16], codes from Loen 2020 [1].

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
