# Peer review of "Constitutionalising the Right to Water in Kenya and Slovenia: Domestic Drivers, Opportunity Structures, and Transnational Norm Entrepreneurs"

_water, doi:10.3390/w13243548_

Round 1
Reviewer 1 Report
The authors present an analysis of the constitutional initiatives in Kenya and Slovenia with respect to the inclusion of a right to water and sanitation. The manuscript is generally well-written, appropriately referenced and annotated. There are a few cases where there is lack of agreement between number:
on line 72, activist should be activists
on line 192, level should be levels
on line 558, activist should be activists
on line 591, suggest should be suggests
on line 776, has should be have.
Please define IO on line 90 (in similar manner as you have defined NGO). Move the definition of GIZ from lines 494-495 to line 390.
On lines 155 and 381, delete the dates: the reference number is sufficient.
On line 198, delete are; on line 251, delete "of the" (the number reference is sufficient).
There are several places where longer quotations are presented in a smaller typeface, which is fine except that these quotations extend the width of the page instead of being indented from the main text. Perhaps these could be set off with quotations (as was done on lines 739-741)?
On line 322, insert "in" before "both": on line 323 perhaps is would be better to indicate that South Africa is "relatively" close to East Africa? South Africa is at the opposite end of the continent. Likewise, on line 387. South Africa is NOT a neighbouring country.
On lines 417-418, the numbers do not add: 84 + 48 do not equal the total of 271 documents. What are the rest?
On line 546, insert a comma between "caves" and "the".
On line 561, delete the full stop after the number 3.
On line 581, change "profile" to "person"--profiles do not speak. On line 603, "talking" probably should be "taking".
On line 631, there seems to be an extraneous number 24. On line 632, insert "During" before "March".
On line 666. delete "does".
With these refinements, the paper should be ready for publication.
Author Response
We want to thank Reviewer 1 for their feedback on our manuscript. We have addressed all the comments and suggestions made by the reviewer. Please see the resubmitted manuscript for reference.
Reviewer 2 Report
Right to water is an important thematic area that you examined in this paper as a comparative perspective. To be considered for further process, you must conduct considerable revisions.
Detailed revisions
- The abstract is not well written. There is no clear entry point and flow of the abstract. Please check this Writing an Abstract for Your Research Paper – The Writing Center – UW–Madison (wisc.edu)
Your start of the introduction must be corrected. Do not start "This article analyses the mobilization......................." Introduction should be given the approachable discussion.
First, you need to clarify why do you select
| Kenya and Slovenia |
as the case studies?
1.1 Line 53, it is not enough just say,
| both countries constitutionalised the right to water |
Line 71 to 73 is not well written and hard to get your real research question.
1.1 I would suggest having a table-based comparison to show the similarities and dissimilarities.
Theoretical framework need to be more widen to say the combination between water quality, sanitation, local engagement and governance. How the different stakeholders compete in different scales in water governance is important to show. This well help to develop to your discussion section.
I suggest reading and cite these papers in a separate section.
Withanachchi, Sisira S., Giorgi Ghambashidze, Ilia Kunchulia, Teo Urushadze, and Angelika Ploeger. 2018. "A Paradigm Shift in Water Quality Governance in a Transitional Context: A Critical Study about the Empowerment of Local Governance in Georgia" Water 10, no. 2: 98. https://doi.org/10.3390/w10020098
Tortajada, C.; Islam, S. Governance in urban water quality and water disasters: A focus on Asia. Water Int.
2011, 36, 764–766
Kayser, G.L.; Amjad, U.; Dalcanale, F.; Bartram, J.; Bentley, M.E. Drinking water quality governance:
A comparative case study of Brazil, Ecuador, and Malawi. Environ. Sci. Policy 2015, 48, 186–195. [
Saravanan, V.S.; McDonald, G.T.; Mollinga, P.P. Critical review of integrated water resources management:
Moving beyond polarised discourse. Nat. Resour. Forum 2009, 33, 76–86.
Pahl-Wostl, C., Gorris, P., Jager, N. et al. Scale-related governance challenges in the water–energy–food nexus: toward a diagnostic approach. Sustain Sci 16, 615–629 (2021). https://doi.org/10.1007/s11625-020-00888-6
Your method and data(it should be materials) cannot be tracked and must be re-written. As your study follow a social sciences approach, you need to read how social sciences illustrates their methods and materials section. It must be followed a clear methodological approach.
Your result section is too detailed. You need to drastically shorten it. Readers, in an academic journal, will not focus detailed historical backgrounds. By writing history and actors and then the regulation system for Kenya and Slovenia is not at all necessary. You must follow a logical argumentation and titles for result section.
Your discussion section also needs to follow a ration combination with the new result section.
Final remarks instead of a conclusion is not advisable. You need to rewrite a conclusion. Overall, your language usage in the paper must be reviewed by a native speaker. There are a lot of syntax and grammar errors.
Author Response
We would like to express our thanks to Reviewer 2. The comments and feedback are have been helpful in our revision of the manuscript.
The abstract have been rewritten to highlight the research question, the theory, method and findings of our study. We have adjusted the start of our introduction as requested by the reviewer, setting off with an approachable discussion of the questions examined in the article.
We thank the reviewer for highlighting the importance of a more thorough discussion of case selection, and we have added a paragraph under 1.1 “Relevance and Rationale” where we clarify the case selection, including a table in which we compare similarities and differences of the two cases. The research question has been made clearer and has been divided into two sub-questions. It has also been moved to the first paragraph of the introduction in order to earlier provide the reader with a clear statement of the aim of the paper.
We have addressed the issue raised by the reviewer regarding theory in the following way: We have included references to the literature on water governance both in the introduction and in the theory section, in order to highlight the importance of water governance structures for the political opportunity structure activists face in a given country.
Regarding the methods section, we have rewritten and edited the methods section to clarify our methodological approach and explain how we have gone about to collect and analyse data. We have also specified “data material” according to the comment made by the reviewer.
In the results section, we have, as suggested by the reviewer cut down on the historical details by omitting the “Brief History”-section and rather included some of this material, as needed in the discussion of the results for each of the cases.
We have also addressed the comment about the final remarks and added a conclusion instead. Please see the resubmitted manuscript for reference.
Reviewer 3 Report
The authors review the process by which water and sanitation was included in the constitutions of Kenya and Slovenia. The paper is generally well-written, with appropriate references. Nevertheless, the authors are encouraged to review the manuscript with particular references to punctuation.
Specific refinements include:
line 19, insert "to" between "order" and "understand"
line 27, insert a comma after "Slovenia"
line 30, delete the comma after "2010"
line 72 (and following), "activist" should be "activists"
line 91, insert a comma after "to"
line 102, define "IOs"
line 147, insert a comma after "cases"
line 157, "normative resources" is used to define "normative dimension"; a better definition should be used rather than defining the term with the term
line 171, delete "of" after "shape"
line 198, delete "are"
line 209. "analyses" should be "analysts": analyses cannot be interviewed.
line 211, delete the comma after "tracing"
line 241. "was" should be "were"
line 251, delete "of the"
lines 241-251, perhaps some mention of the several UN decades for water and water and sanitation would be appropriate here. While these initiatives were not conventions in the sense used by the authors, they did represent international consensus on the importance of water and sanitation to human health and well-being
line 277-278 and following: these quotation should be indented (the margins need to be changed in each case where extensive quotations are included in the manuscript)
line 300, is the "Independent Expert" and the "Special Rapporteur" the same person? It would be helpful to give a name. Insert "the" in front of "Special"
line 307, move reference [25] to line 305 after "Baer"
line 318, insert "the" before "people"
line 323 and following (line 387), South Africa is at the opposite end of the continent from Kenya (East Africa); to use words like proximity is misleading.
line 342, "maintained" would probably be better stated as "extended"
lines 358-360, the verb forms should be changed to "identifying", "collecting", "conducting", "proposing" and "creating" for consistency.
line 390, define the acronym "GIZ"; move the definition from line 494
line 412, "define" should be "defined"
line 413, delete the comma after "prevailed"
line 415 and following (lines 424 and 432, for example), "right" should be "rights"
lines 417-418, N = 132, not 271--what are the other keywords?
line 455, insert a comma after "and"
line 464, define the acronym "CKRC"
line 511, "educate" should be "inform" (unless you mean classroom-based teaching)
line 521, a reference would seem to be needed after the statement regarding Iceland and Hungary
line 538, insert a comma after "but"
line 540, insert a comma after "and"
lines 551 and 553, "suppliers" may be a better word choice
line 552, insert a comma after "company"
line 558, "activist" should be "activists"
line 563, insert a comma after "already"
line 584, "actor" should be "factor"
line 588, "opportunities of" should be "opportunities for"
line 590, "discuss" would be better stated as "encourage discussion of"
line 591, delete the comma before "suggest"; "suggest" should be "suggests"
line 631, to what does the number "24" refer? It probably should be deleted.
line 632, insert "In" before "March 2014"
line 647, "structures" should be "structure" OR "is" should be "are" (there should be agreement between subject and verb forms)
line 655, "good" should be "goods"
line 666, delete "does"
line 667, "efforts of" should be "efforts for"
line 711, "use" should be "used"
line 720, "responsibility of" should be "responsibility for"
line 727, "sign of" should be "sign off"
line 730, a reference would seem to be required for the Sustainable Development Goals
line 746, insert a comma after the second "and" (in front of "more specifically")
line 776, "has" should be "have"
line 777, "are" should be "were"
With these largely editorial refinements, the manuscript should be ready for publication.
Author Response
We want to thank Reviewer 3 for their feedback on our manuscript. We have addressed all the comments and suggestions made by the reviewer. Please see the resubmitted manuscript for reference.
Round 2
Reviewer 2 Report
You have conducted the revisions accordingly to the comments. I accept the manuscript for the publication process.